# Endocrine Disruptor Chemicals and Children’s Health

**DOI:** 10.3390/ijms24032671

**Published:** 2023-01-31

**Authors:** Giada Di Pietro, Francesca Forcucci, Francesco Chiarelli

**Affiliations:** Department of Pediatrics, University of Chieti “G. d’Annunzio”, 66100 Chieti, Italy

**Keywords:** endocrine-disrupting chemicals, endocrine disruptors, COVID-19, endocrinopathies, pediatrics

## Abstract

We are all exposed to endocrine-disrupting chemicals (EDCs) starting from embryonic life. The fetus and child set up crucial developmental processes allowing adaptation to the environment throughout life: they are extremely sensitive to very low doses of hormones and EDCs because they are developing organisms. Considering the developmental origin of well-being and diseases, every adult organism expresses consequences of the environment in which it developed. The molecular mechanisms through which the main EDCs manifest their effects and their potential association with endocrine disorders, such as diabetes, obesity, thyroid disease and alteration of adrenal hormones, will be reviewed here. Despite 40 years having passed since the first study on EDCs, little is yet known about them; therefore, our purpose is to take stock of the situation to establish a starting point for further studies. Since there is plenty of evidence showing that exposure to EDCs may adversely impact the health of adults and children through altered endocrine function—suggesting their link to endocrinopathies—it is essential in this context to bear in mind what is already known about endocrine disruptors and to deepen our knowledge to establish rules of conduct aimed at limiting exposure to EDCs’ negative effects. Considering that during the COVID-19 pandemic an increase in endocrine disruptor effects has been reported, it will also be useful to address this new phenomenon for better understanding its basis and limiting its consequences.

## 1. Introduction

Endocrine-disrupting chemicals (EDCs) are exogenous agents that interfere with normal endocrine physiology by influencing hormone synthesis, metabolism, and/or cellular actions. EDCs’ major source results from industrial processes, but they can be present naturally in soy, legumes and other plant-based products; they can be easily found in air, water, soil, a large variety of household products and medical devices (clothes, drugs, medical devices, sanitizers, food and its containers, furniture, cosmetics, personal care tools, toys, construction materials and so on), thus becoming ubiquitous in our environment [1]. As with other environmental contaminants, these substances can cross the placenta and their role in the developmental origin of diseases such as obesity and diabetes has been proven. Epidemiological data suggest that the rise in diabetes, cancer and infertility in the past two to three decades could be attributable, at least in part, to in utero exposure to EDCs, and there is evidence from epidemiological case studies and complementary animal- and cell-based models that exposure to endocrine-disrupting chemicals, both in utero and during one’s lifetime, can have effects on human health [2].

EDCs may affect the organism by means of direct interaction with hormonal receptors or by influencing enzymatic stages of steroidogenesis and neurotransmitter synthesis; they might also affect epigenetic regulation of endocrine and nervous pathways.

The molecular mechanisms of EDCs’ action and their potential association with endocrine disorders will be reviewed here.

## 2. Endocrine Disruptors—Defining Criteria

Since the term “endocrine disruptors” was first used, numerous definitions have been proposed by various groups and agencies. The 2002 report by the International Program on Chemical Safety and World Health Organization (IPCS/WHO) defined an EDC as “an exogenous substance or mixture that alters the function of the endocrine systems and consequently causes adverse effects in an intact organism, or its progeny, or subpopulations”. This definition was used again in the 2012 State of Science on EDCs published by the UN Environment Program (UNEP) and WHO. Using the major components of the IPCS/WHO and the UNEP/WHO definition of an EDC [3], identifying a compound as an EDC, therefore, requires an appraisal of the following:Evidence of an (adverse) effect (remembering that reversible effects might be adverse, that there can be a continuum of effects from “initiating events” to “apical effects” induced by the chemical, and that there remains a debate about what should be considered “adverse” outcomes) [4,5,6].Evidence of endocrine-disrupting activity (remembering that endocrine-disrupting activity extends beyond “endocrine active” compounds and includes disruption to hormone binding, synthesis, secretion, transport and metabolism).Evidence of a plausible link between the observed adverse effect and the endocrine-disrupting activity.

Knowing the methodological limitations in assessing consolidated and universally effective conclusions for what EDCs are concerned, when talking about EDCs it is important to focus on longitudinal studies (since exposure to EDCs may vary over time) and at the same time, to consider the various types of exposure with particular focus on the type of food, the environment and the lifestyles of the subjects studied. To try to identify sub-groups in the various populations that are more exposed or more sensitive to the toxic action of EDCs (by sex or ethnicity or by locating them in different geographical areas) is also a main goal along with the purpose of considering the role not only of the individual EDC but, as is generally the case in “real life”, of mixtures of substances. 

It is also relevant to standardize laboratory and statistical analysis methodologies in order to increase the accuracy of the evaluation, not only of the presence and concentration of EDCs, but also of their biological action through the modern techniques of the so-called “omics” (proteomics, metabolomics, transcriptomics, etc.) trying to converge the data of the various “omics” with those of genomics to reach the desirable personalized medicine in order to achieve a deepened knowledge of epigenetic interference [7]. 

## 3. Differences between Adults and Infants

Infants and children have multiple differences from adults and that might significantly influence their response to EDCs in terms of higher exposure and, therefore, more evident negative outcomes; this mismatch is due to differences primarily in physiology and anatomy, but also in pharmacokinetics, diet and behavior [8,9,10]. Children are also more sensitive to the effects of EDCs than adults: the different kinetics of environmental chemical metabolites often result in higher concentrations of EDCs (in circulating blood or tissues) for an administered dose. During early development, many factors are programmed and most of them are sensitive to disruption by means of EDCs, resulting in an increased risk of developing childhood diseases; this means that early-life exposure to EDCs can result in promoting childhood obesity, liver dysfunction and cardiometabolic impairment by perturbing the neuroendocrine system. To be specific, the main differences may be summarized as follows: -The amount of water, food and air introduced into children’s organisms in proportion to body surface area (BSA) outclasses the one needed in adults.-The immaturity of children’s blood–brain barrier makes them more sensitive to neurological damage.-Infants’ skin is more water-permeable.-Children spend more time inside buildings and settings rich in sources of EDCs, such as construction materials but also tools of everyday use and toys; in addition, infants’ tendency to mouthing increases their exposure to EDCs.-During developmental age, biological systems and organs are in different stages of maturity and functionality, and this makes the detoxification system less efficient.

These perturbations (which appear to be more significant in children) may lead to an altered glucose metabolism by means of metabolic changes following one another in a cascade; this consequently increases pancreatic insulin secretion and resistance, resulting in a significantly reduced tissue response to insulin-mediated cellular actions, reducing the effectiveness of insulin in stimulating glucose usage and suppressing hepatic glucose output. In addition, it accounts for interfering with insulin function on the metabolism of lipids and protein; it is to be taken into consideration that the gene expression and function of the vascular endothelium are also affected.

## 4. Routes of Exposure, Absorption, Secretion

The extensive use of EDCs in multiple fields justifies the wide dissemination of EDCs into the environment (in the atmosphere, in waste disposal, in car emission and in water by sewage).

Considering the food chain, it is possible to divide EDCs into four categories [11]: EDCs which might end up in bioaccumulation, mostly compounds with adipose deposition; they can be found in animal fatty acids such as milk and its derivatives (e.g., dioxin, polychlorinated biphenyls (PCBs), polybrominated diphenyl ethers (PBDEs), and perfluorinated compounds).EDCs used for food-producing mechanisms (FCMs): examples are pesticide, food additives or OGM; at present time, the research is working on identifying proper markers in order to detect the presence of EDCs in food and to protect the consumer.EDCs released into food products coming from tools used for food storage, transportation or other industrial processes; among them, bisphenol A must be taken into account, since its use in food packaging has been restricted in the EU (even though not prohibited) since 2011 [12].Natural EDCs with hormonal-like actions. Chemical structure, dosage and cumulative duration of exposure, as well as the hormonal status of the exposed organism, are all directly determining factors of EDCs’ action. Phytoestrogens, quercetin, iodine, or other heavy metals (so-called metalloestrogens) are some examples of these kinds of EDCs.

EDCs affect individuals by multiple mechanisms: inhalation, ingestion and direct cutaneous contact. The first two are the most associated with a significant level of absorption and, therefore, with a higher level of circulating EDCs; nevertheless, thanks to hepatic enzyme action, those are also the means to better excretion and detoxification if compared with cutaneous direct contact. At variance, the cutaneous absorption causes catabolism which is exclusively in situ.

It must be taken into account that there might be a coexistence of more than one endocrine disruptor in the same food, resulting in a synergistic/additive effect. Moreover, once in the body, EDCs tend to produce bioaccumulation phenomena, especially in the adipose tissue; this condition is also known as the “cocktail effect” [13,14].

## 5. Main Endocrine Disruptors

We discuss below the characteristics of the main endocrine disruptors (summarized in Table 1).

Phthalates: These are the esters of 1,2-dibenzene dicarboxylic acid. Phthalates are a class of EDCs used in several consumer products because of their effect in making plastics softer, more flexible and expandable; examples might be personal care items, medications, building materials, pesticides, gastro-resistant drug linings, and perfumes. Biomonitoring studies from all over the world indicate that there is universal phthalate exposure among pregnant women, infants and children. Phthalate exposure occurs through ingestion, inhalation or dermal absorption. It has been shown that phthalates can cross the placenta, resulting in exposure to the fetus [15,16,17,18,19]. 

The hydrolyzation of phthalates (to their respective mono-ester metabolites) occurs rapidly after inhalation or ingestion; LMW (low molecular weight) phthalates (di-ethyl phthalate, di-n-butyl phthalate, and di-iso-butyl phthalate) are metabolized into glucuronide or sulfate-conjugated hydrolytic monoesters and eliminated by the kidneys. However, mono-2-Ethylhexyl phthalate, which is the of di-2-Ethylhexyl phthalate’s (DEHP) metabolite, undergoes additional enzymatic oxidation; subsequently, it is excreted and conjugated. Given the shortness of their biological half-lives (<24 h), phthalates persist in the body for a short time; nevertheless, there is persistent exposure to them. Their predominant excretion in urine and their considerably lower blood levels may be subject to exogenous contamination, invalidating sample processing, storage or collection; that is to say, assessing exposure using urine biospecimens might be more accurate and it requires the collection and analysis of multiple samples. 

As observed in a recent investigation conducted in Luxembourg [20], hair might be a better-suited matrix for this kind of exposure assessment and is gaining increasing attention for this purpose in epidemiological studies. Compared to urine in which only hydrophilic metabolites can usually be detected, hair is suitable for assessing exposure to many different families, including parent compounds and metabolites of both persistent and non-persistent organic pollutants, being at the same time representative of wider time windows, covering up to several months depending on hair length. In contrast, urine only gives information on short-term exposure (and present high variability): a 6-cm-long hair sample provides information integrating exposure over 6 months; approximately 50 urine samples collected over the same period only represent 3–4% of the total volume of urine produced over the same period.

The action or metabolism of glucocorticoids, thyroid hormones and androgens seem to be the most affected by phthalates interference [21].

PFAS (Per- and poly- fluoroalkyl substances): These substances are a class of engineered fluorinated chemicals largely produced in the 1950s. The most used PFAs are PFOA (perfluorooctanoic acid) and PFOS (perfluorooctanoic sulfonate). They can be found in a wide range of products in common use, such as food containers and water-resistant coatings for textiles, floor polish, industrial surfactants and even fire-fighting foam. PFAS are hallmarked by an alkyl chain attached to multiple fluorine atoms. PFAS are extremely resistant to thermal, chemical and biological degradation, thanks to the strong carbon–fluorine bond, resulting in persistence in human tissues for years by means of bioaccumulation. These compounds have long biological half-lives (3.8 to 7.3 years) and some of them (perfluorooctane sulfonate, perfluorooctanoic acid, perfluorononanoic acid, and perfluorohexane sulfonate) can cross the placenta [22].

Bisphenol A (BPA) [23,24,25]: BPA is a compound belonging to the group of diphenylmethane derivatives and bisphenols (synthetic, organic); it is one of the most investigated substances for its endocrine disruptor properties and at the same time is in the center of many EDC-related controversies. The analysis on how BPA fits into the regulatory identification as an EDC is a challenge in terms of methodology [26]. 

BPA is involved in the production of polycarbonate plastics, resins and PVC; these materials are involved in the production of a wide range of objects in common use, such as containers to cook and preserve food, cans, drinking-water dispensers, cutlery and plates, but they can also be found in mobile phones, toys, cables, tires, packaging and electronic devices. The main exposure route to BPA is oral ingestion because of its involvement in containers for food and beverage; additional routes of exposure among people using materials containing BPA are dermal absorption and inhalation. Bisphenol A is excreted in the form of glucuronide-sulfate conjugates in the urine; however, having an estimated 6 h biological half-life, it does not stay in the body for long. Since BPA is almost exclusively excreted in the urine, to assess its exposure we have to measure urine concentrations of BPA and of its conjugates. Biomonitoring studies worldwide indicate nearly universal BPA exposure among pregnant women, infants and children. Current safety assessments by a preponderance of international regulatory agencies (Food and Drug Administration, European Food Safety Authority—2014, 2015) conclude that BPA does not threaten humans at current levels of dietary exposure at any life stage; nevertheless, BPA is likely to be a human health hazard according to multiple studies.

Triclosan [27,28]: This is a chlorinated aromatic compound endowed with functional groups representative of both phenols and ethers. It is a chemical compound with antimicrobial properties disrupting cell membrane integrity and bacterial lipid synthesis; it can be found in many consumer products. Exposure to triclosan occurs mainly through dermal and oral absorption. It is not persistent in the body, has a biological half-life of <24 h, and is mainly excreted in the urine as a sulfate conjugate or glucuronide; for the same reasons as BPA and phthalates, triclosan exposure is well quantified using urine specimens. Studies of biomonitoring suggest that exposure to this compound is universal among children and pregnant women.

**Table 1 ijms-24-02671-t001:** Main endocrine disruptors, their possible effects on human body and main routes of exposure. (GDM = Gestational Diabetes Mellitus; PCOS: polycystic ovarian syndrome; IUGR: intrauterine growth restriction) [29,30].

Substance	Routes of Exposure	Probable Effects
Phthalates	Ingestion, inhalation, dermal absorpiton. Fetus exposed (this EDC can cross the placenta).	Interfere with action/metabolism of androgens, thyroid hormones, glucocorticoids.
PFAS	Bioaccumulation (through food and textiles. Fetus exposed (this EDC can cross the placenta).	Impaired fasting gucose/glucose tolerance, obesity, PCOS, IUGR, GDM.
BPA	Ingestion, inhalation, dermal absorpiton.	DMT2, obesity in infancy, impaired glucose tolerance.
Triclosan	Ingestion, dermal absorpiton.	Precocious puberty, rapidly progressive puberty, precocious menarche.

## 6. Mechanism of Interference

EDCs can act through several different pathways either singly or in combination to alter the function of cells (Figure 1):

Classic nuclear receptor pathway—The most established cellular effects of EDCs are their ability to activate or inhibit nuclear receptor signaling. Nuclear hormone receptors are a family of ligand-modulated transcription factors that manifest their biological effects by interacting with DNA response elements within target genes [31]. The receptors recruit coactivator or corepressor proteins to the DNA response elements, enabling them to activate or repress gene expression [32]. EDCs can bind to the ligand-binding pockets of hormone receptors for estrogens (ERs), androgens (ARs), progestins (PR) and thyroid hormone (TR) to mimic or disrupt their cellular activities [33]. In addition, the glucocorticoid/mineralocorticoid (GR/MR) system can be identified as an EDC target [34]. Many EDCs bind with significantly (>1000-fold) lower affinities [35]. The combination of relatively low binding affinities of EDCs for hormone receptors and the relatively high doses required for activation [36] suggest that the adverse effects of these agents may occur through alternative signaling pathways.

Nongenomic nuclear receptor signaling—EDCs that mimic estrogen action can also activate nongenomic signaling through the seven transmembrane ER, GPR30. Genistein and bisphenol A (BPA) possess considerable relative binding affinities for GPR30 compared with estradiol [37], and both activate GPR30 signaling with similar efficacy as estradiol. Doses between 1 pM and 1 nM of BPA and other EDCs are sufficient to elicit biological estrogenic effects [38]. BPA has been detected at nM concentrations in human adult and fetal serum, breast milk, amniotic fluid and cord blood [39]. These levels are therefore sufficient to activate GPR30, and it is likely that the estrogenic effects of BPA are mediated through nongenomic ER signaling rather than through the classical pathway [40]. This mechanism may also be relevant with EDCs that affect other nuclear receptors.

Epigenetic effects—Developmental reprogramming refers to alterations in the epigenome that occur during the prenatal and early postnatal periods. This is a critical window when epigenetic modifications such as the methylation of histones and cytosine-guanine sequence [CpG] sites in DNA are made on the genome [41]. There is now evidence that early exposure to EDCs may induce developmental reprogramming in the adult, with a potential to transmit it to future generations. It is known that the use of diethylstilbestrol (DES) during pregnancy resulted in significant reproductive abnormalities and rare cancers in offspring [42]. There is also emerging evidence that early exposure to BPA and other EDCs may increase susceptibility to chronic diseases in adulthood, including endocrine disorders, diabetes and cancer [43]. One report described the ability of EDCs, through induction of nongenomic signaling, to activate a histone methyltransferase and alter histone methylation patterns of genes associated with prostate cancer. These events occurred during prostate development, and these epigenetic modifications persisted throughout adulthood where they resulted in elevated basal and hormone-induced expression of reprogrammed genes [44].

Epidemiological studies have found an association between EDCs exposure and obesity. Experimental data suggest that polycyclic aromatic hydrocarbon (PAH) exposure in utero results in altered methylation and enhanced activity of the peroxisome proliferator-activated receptor-gamma, the primary regulator of adipogenesis [45], providing a mechanistic explanation for the epidemiologic findings.

## 7. EDCs, Diabetes and Obesity

The latest pieces of evidence concerning EDCs’ role in the development of metabolic dysfunctions and obesity have shown some interesting results. The Parma consensus statement (2014) [46] hypothesized that many of the pollutants cited above may induce alteration in metabolic processes such as obesity and insulin resistance (IR) in humans. EDCs also have a role in the manifestations of metabolic syndrome through inflammatory processes via cytokines/adipokines, resulting in metabolic imbalance leading to IR both through an obesity-inducing mechanism and direct action on the pancreatic beta cells [30]. Environmental pollutants can affect multiple aspects of β-cell physiology, including β-cell function and survival, insulin release and glucose provision. BPA can cause weight gain, glucose intolerance, diabetes mellitus type 2 (T2DM) and fatty liver in mice. We will focus on EDCs for which there is widespread general population exposure.

According to the diabetogenic hypothesis [47], EDCs exert their action either by disturbing insulin sensitivity in peripheral tissues or impairing insulin production in pancreatic beta cells, behaving as diabetogenic substances. A study published in June 2022 [48] demonstrated the proapoptotic effect of BPA on pancreatic β cells: it interacts with the crosstalk among three estrogen receptors (ERα, ERβ and GPER). Activation of GPER by G1- or BPA-induced apoptosis via ERα and ERβ; BPA directly decreases ERαβ heterodimers via ERα and ERβ or after activation of GPER. This decrease in ERαβ heterodimers disrupts the active antiapoptotic effect of ERα and ERβ in beta cells. A meta-analysis of prospective and cross-sectional studies concluded that there might be a statistically significant relationship between PCBs, levels of dioxins, BPA, organochloride pesticides and diabetes prevalence; [49] meanwhile, a cross-sectional study conducd in adults confirmed the relationship between diabetes and PFAS [50]. Prospective evidence also points out the associations between background exposures to PCBs, organochlorine pesticides and DM (diabetes mellitus), and demonstrates a relationship between BPA, dioxin and prevalence of DM. Various molecular patterns involved in obesogenic conditions might be responsible of IR provocation: EDCs are also responsible for interfering with hormones and other endocrine factors such as adiponectin, resistin, leptin, and adipsin [51,52]. The cause of IR can be identified with the inhibition of GLUT4 caused by excessive expression of RBP-4 (retinol binding protein-4) in abnormal adipose tissue [50]. As this occurs, the metabolic state of increased fasting glucose induces hyperinsulinemia, which in turn stimulates transcription factors in the liver, driving hypertriglyceridemia and hepatic steatosis. EDCs cause intrauterine growth retardation by means of confining essential metabolic substrates to the fetus, presenting as fetal starvation and mimicking the metabolic basis which triggers diabetes progression in PDX-1. It is reasonable to believe that prenatal and early-life exposure to PCBs, BPA, dioxins and perfluorinated compounds negatively affect the immune system development, resulting in immune disorders such as type 1 DM. In addition, epigenetic and hormonal alteration are involved in the way that BPA and PCB deregulate development of peripheral IR, insulin production, pancreatic islet beta-cell function and mass, impairing insulin signaling, output and increasing cell apoptosis; by these means, those EDCs promote DM onset in insulin-resistant, obese individuals with type 2 DM; as an example, PCB activates the aryl hydrocarbon receptor (AHR) and inhibits the transcription of factor Nrf2a and PFOA inducing pre-proinsulin expression by increasing the proinsulin/insulin ratio. The EDCs listed above, by different mechanisms, lead to an increase of type 2 DM through modulation of glucose metabolism [30]: PCBs and organochlorines (OCs) act through endocrine-disrupting mechanisms and mitochondrial dysfunction, including PCBs’ effects on pancreatic beta-cell function and OCs adiponectin release. EDCs decrease GLP-1R (glucagon-like peptide 1 receptor), which increases the release of pancreatic glucagon via hypothalamic receptors as a lack of satiety during eating. EDCs play a key role in obesity-associated IR due to the activation of the extracellular matrix receptor pathways in adipose tissue that constitute the cell microenvironment. EDCs contribute to apoptosis, inflammation and angiogenesis in adipose tissue, liver and skeletal muscle, playing a role in extracellular matrix remodeling through its receptors such as integrins and CD44. These compounds also affect quantitative insulin secretion, also altering insulin-dependent mRNA stability. Since the insulin-like growth factor-binding protein (IGFBP-1) gene promoter regulates blood glucose levels, the specific upregulation of IGFBP-1 mRNA might account for the disruptive effects of tetrachlorodibenzo-p-dioxin (TCDD) on glucose metabolism in human hepatocytes and HepG2 human hepatoma cells. TCDD impairs insulin secretion leading to cellular insulin reservoir consumption and reduces glucose uptake in the pancreas, suggesting that insulin deficiency may ensue after sustained exposure to this compound.

Intrauterine growth retardation and the subsequent occurrence of IR seems to be the result of a deleterious fetal nutritional environment produced by diethylstilbestrol (DES) exposure [53]. The adverse endocrine disruptive effects of BPA as a disrupter of pancreatic beta cells, as well as on the endocrine system, are suggested by several studies: studies on animals concluded that pregnant mice treated with BPA during gestation, at environmentally relevant doses, exhibit altered insulin sensitivity and significant glucose intolerance, thus becoming, several months after delivery, overweight, mainly through impairments in beta-cell function and mass [54]. In vivo experiments have been conducted, showing an effect of BPA exposure on insulin release (increased) and glucose-stimulated insulin secretion in an estrogen receptor-a (ERa) dependent model [55].

Sex steroids exert important effects on metabolic target tissues, including the pancreas, checking β-cell insulin secretion in both cGMP-dependent and independent pathways. Several EDCs, including arsenic, DEHP and BPA, are known for being β-cell function disruptors, promoting oxidative stress [56]. Since pancreatic β cells are innately more sensitive, oxidative stress is responsible for significantly compromising β-cell function; given that assumption, it can be concluded that BPA accelerates the exhaustion of β-cell reserve via immune modulations in pancreatic islets. In this perspective, it is quite obvious that the immunomodulatory effects of BPA in this animal model lead to the conclusion that EDCs might also contribute to the increasing Diabetes Mellitus type 1 (DMT1). However, the current evidence has not yet established a certain association between EDCs and metabolic alterations, especially in developmental age; therefore, more prospective studies are paramount [57].

## 8. EDCs and Thyroid

A normal thyroid function plays an extremely important role in the growth, development and regulation of energy metabolism and the immune system. The production of thyroid hormones is dependent on a complex network of mechanisms involving the hypothalamic–pituitary–thyroid axis and a number of fine mechanisms aimed at the regulation of the iodine–tyrosine combination. [58]. EDCs widely affect this process.

Since thyroid hormones are essential for normal brain and skeleton development, low concentrations of thyroid hormones are associated with growth retardation and neuromotor and psycho-behavioral development alteration that might determine cretinism [59]. Given its complexity, the thyroid hormonal system has numerous sites of potential dysregulation by means of multiple EDCs. The result of these interferences may determine a reduction in T4 and an increase in TSH leading to hypertrophy and consequently to hyperplasia of the thyroid follicles resulting in an increased thyroid volume. In the long run, the constant stimulation of follicular cells might lead to the cancerous transformation of cells and the evolution toward follicular thyroid cancer.

The large variety of studies about early life exposure to EDCs during pregnancy and in newborns concluded that it is extremely relevant. A recent Danish study involving a group of pregnant women suggested an effect of persistent organic pollutants (EDCs) exposure in utero on placental thyroid hormones [60,61]. Data from studies based on the measurement of peripheral levels of thyroid hormones show effects on growth and development confirming the hypothesis of a negative EDCs effect on thyroid functioning [62].

Moreover, a study [63] (conducted on 170 healthy mother–infant pairs) concluded that PCB levels in 30-months-old infants correlated with their mental development index; the Hokkaido study [64], instead, demonstrated that PCB exposure during pregnancy results in an increase in thyroid hormones both in mothers and newborns. The Tainan birth cohort study showed that prenatal phthalate exposure during gestation altered cord and serum thyroid hormone homeostasis [65]; this was also confirmed in a large cohort in the United States [66], exposed to multiple phthalate metabolites during pregnancy. In a large group of Norwegian mother–child pairs, a cohort study was conducted examining the relationship between the levels of prenatal phthalate exposure, maternal thyroid function, and the risk of attention deficit and hyperactivity disorder [67]; the study concluded that the higher the exposure is, the higher the risk of developing attention deficit hyperactivity disorder (ADHD). Recent findings from a cohort study [68] (pregnant women in whom BPA urine levels and serum thyroid hormones were measured during pregnancy) showed the dependence of the relationship between thyroid function and BPA on gestational age: a negative association was shown in early gestation. Although prenatal and early exposure to pollutants has received the largest attention from the scientific community, some studies have also examined children and young adults.

A study, aimed at exploring the negative association with free thyroid hormone levels, was conducted on children living in a rural area in Brazil; blood levels of metals (chromium, manganese, mercury, lead) as well as hair EDC levels were examined [69]. The data from epidemiological studies evaluating the association between thyroid alterations and pollutants is still debated; the difficulty is in obtaining homogenous populations and a complete panel of pollutants.

The Horizon 2020 project, therefore, invested in this particular field by financing the ATHENA project: Assays for the identification of Thyroid Hormone axis-disrupting chemicals, elaborating Novel Assessment strategies [70].

The ATHENA consortium aims to mobilize the scientific progress made in recent years to close critical gaps left open in test methods for thyroid hormone axis disrupting-chemicals in order to develop new methods for incorporation into existing OECD test guidelines that can capture the consequences of maternal thyroid hormone deficiency on the developing brain, due to the disruption of delivery of thyroid hormones to the fetus. To make this possible, research aims to establish new endpoints for identifying down-stream effects on the developing brain in fetal and post-natal life. The purpose is to provide new test methods for chemicals that interfere with the delivery of thyroid hormones to the fetus and the adult brain across physiological barriers such as the placenta, the blood–brain barrier and the blood–brain cerebrospinal fluid barrier [71].

## 9. EDCs and Adrenal Hormones

In the modern era, puberty has occurred earlier. Indeed, the average age of menarche increased from 17 years in the early 19th century to 13 years in the mid-20th century [72].

Since the variation of the onset of puberty is determined by non-genetic factors for a variable percentage between 20% and 40%, it is reasonable to think that among the key factors involved in its regulation, an important role is played by EDC [73]. Prenatal, postnatal life and puberty are the three main temporal windows of susceptibility when EDCs may act.

There is some evidence in the literature that shows that the increase in precocious puberty may also be partly caused by exposure to EDCs [74]: in the early 1980s, a progressive trend toward premature thelarche was noted in Puerto Rico, where higher serum phthalate levels were demonstrated in Puerto Rican girls with premature thelarche [75].

### 9.1. Genital Development

Exposure to hormone mimics during development can have health repercussions that may become apparent at birth or manifest later in life [76].

Adequate sex steroid hormone concentrations are essential for a normal fetal genital development in early pregnancy and soon after birth. The hypothalamic–pituitary–gonadal (HPG) axis is briefly activated during the first six months of postnatal life, a period known as “mini-puberty”, during which increased sex hormone concentrations are found. Thereafter, the HPG axis remains inactive until pre-puberty [77]. It could be a useful “time window” for a very early diagnosis of hormonal deficits related to the sexual sphere that clinically manifest many years later. However, it also represents a crucial period of susceptibility to EDCs.

Some EDCs have shown antiandrogenic effects or act as androgen receptor agonists, which means that they may suppress or inhibit the biological effects of androgens and the normal body tissue response to these hormones. In particular, the downregulation of fetal testosterone production may alter reproductive tract development.

Particularly, many studies have examined the relationship between EDC and anogenital distance (AGD), the distance between the anus and the genitalia (penis in boys, clitoris in girls), which is hypothesized to reflect the androgenicity of the uterine environment and, therefore, can be used as a predictor of subsequent adverse health effects from prenatal exposure to EDCs.

In boys, most studies of phthalates of both high and low molecular weight measured in prenatal urine (*n* = 8) or umbilical cord blood (*n* = 1) reported associations with shorter anogenital distance (a feminizing effect) or lower anogenital index [78,79,80,81,82]. In girls, there was not a clear association with exposure to EDCs during pregnancy: results for bisphenol A (BPA) were inconsistent, and there was too little evidence regarding triclosan, PFAS and PBDEs. Recently, Jensen et al. [83] studied the associations between maternal paraben concentrations in second trimester urine and AGD and reproductive hormone concentrations at 3 months of age in offspring.

Parabens are weak estrogens capable of inhibiting the metabolism of endogenous estrogens which in turn deactivate the HPG-axis leading to lower FSH, LH and steroid concentrations [84]. Higher maternal paraben exposure was associated with shorter AGD in male offspring and lower concentrations of reproductive hormones (FSH, LH, DHEAS, 17-OHP) in girls. However, parabens are quickly metabolized with urinary excretion half-life of less than 24 h [85]; therefore, a single spot-urine sample collected around gestational week 28 may not reflect fetal exposure in the sensitive developmental window in early gestational age or during mini-puberty. The long-term effects of changes in reproductive hormone concentrations during mini-puberty have not yet been investigated. Further studies on the effects of both AGD and childhood reproductive hormone concentrations on future reproductive health are needed [54].

### 9.2. Implications othe Female Reproductive System

Early life exposure to EDCs, especially during prenatal and early postnatal development, alters development of the female reproductive system [86,87,88,89].

Until the ban on bisphenol A (BPA) by various regulatory agencies [90], BPA was a component of baby bottles, pacifiers, baby food packaging and toys for babies and children. The actual impact of early exposure to bisphenol A (BPA) is not yet known as epidemiological data are lagging behind, but animal data suggest that exposure to BPA at environmentally relevant doses is associated with ovarian cysts, proliferative lesions in the oviduct, cervical sarcoma, uterine polyps and breast adenocarcinoma [91]. The worrying relevance of these EDCs is, therefore, justified.

### 9.3. Implications on the Male Reproductive System

Reports of deteriorating male reproductive health are well known. Serum testosterone levels and sperm quality are declining [92,93,94].

Testicular dysgenesis syndrome (TDS) refers to the effects of impaired testicular embryonic development caused by a combination of genetic and environmental factors such as maternal exposure to EDCs [95,96]. Indeed, exposure to EDCs appears to be associated with disorders manifested at birth, such as congenital cryptorchidism, a congenital penile anomaly called hypospadias, and reduced anogenital distance (AGD), as well as disorders presented later in life, including poor quality of sperm, testicular germ cell tumors and altered levels of reproductive hormones.

Uterine exposure to antiandrogen phthalates during the male programming window has been shown to cause a direct testicular toxic effect resulting in decreased testosterone production and, as a result, leads to an increased risk of developing hypospadias and cryptorchidism. A large Canadian study measuring EDCs, specifically polybrominated diphenyl ethers (PBDEs), in hair samples obtained from women within 18 months postpartum, reported a positive association with cryptorchidism [97].

## 10. Effects on Neurodevelopment

Prenatal exposure to EDCs can affect fetal neurodevelopment resulting in attention deficit disorder (ADHD), autism spectrum disorder, and cognitive and behavioral dysfunction that can manifest from infancy to adulthood, permanently.

This happens because the developing nervous system is particularly sensitive to environmental stresses, as it is going through several critical evolutionary processes (i.e., neuronal proliferation, migration, differentiation, synaptogenesis and myelination).

Specifically, the most susceptible time window, known as the brain growth spurt (BGS) period, begins during the third trimester of pregnancy and continues through the first two years of life.

At the basis of this phenomenon there are two distinct pathways: the alteration of the function of the sex hormones resulting in dimorphic brain development, and the disturbance of the maternal thyroid function [98,99,100].

Thyroxine (T4) and free T4 (FT4) were negatively associated with urinary levels of DBP; therefore, exposure to dibutyl phthalates (DBP) seems to interfere with the thyroid function of pregnant women [98].

Experimental data showed that an early exposure to phthalates may disrupt structure and function of the hippocampus [99], a region of the brain associated with internalizing behaviors, such as anxiety and depression. Furthermore, Daniel et al. [101] suggested a sex-specific effect on children’s behavior after exposure to phthalates during prenatal life and early childhood: boys exposed to non-DEHP phthalates tended to have anxious–shy behavior while girls exposed to DEHP metabolites showed hyperactivity.

Six studies showed decreases in IQ at the age of 6–10 years old in children exposed in prenatal and perinatal periods to organophosphate pesticides and PBDEs [101,102,103,104,105].

The association with autism spectrum disorders (ASDs) and EDCs has been investigated in relation to exposure to phthalates [101], but the increased risk of developing ASDs has been demonstrated especially with exposure to organophosphate pesticides through studies conducted in California [106,107,108,109,110], New York State [87] and Cincinnati [111] based on pesticide use registries and urinary concentrations of their metabolites.

## 11. EDCs and the COVID-19 Era: How Did the Pandemic Affect the Phenomenon?

A significant increase in precocious puberty has been reported in Italy since the initial lockdown due to the pandemic. The mechanism underlying this phenomenon is not yet known with certainty. Changes in lifestyle, screen time and sleep habits were initially thought to be to blame, but these factors do not seem sufficient. Probably, changes in central nervous mediators have occurred but these hypotheses have yet to be investigated.

Of course, spending more time at home and increasing the use of disinfectants has certainly exposed these children more to EDCs.

Since 2020, a massive use of disinfectants has been observed among adults and children. Currently, there are many commercial products containing antimicrobial agents and the effect of the constant use of these products on human health is not yet known. Triclosan deserves a special mention as it represents the main antimicrobial agent used in cosmetic products, toothpastes and disinfectants: its greater use in our daily routine can significantly interfere with hormonal function, its estrogenic and androgenic activity being already known [112,113]. It seems mandatory to investigate this phenomenon that has occurred on a global scale with appropriate studies to verify what is happening and to obtain new information on the consequences of the COVID-19 pandemic and on precocious puberty for future prevention.

EDCs are even hypothesized to increase susceptibility to the risk of severe COVID-19. If the COVID-19 pandemic has influenced EDCs’ exposure and therefore their effect, it is also true otherwise: several environmental contaminants have been implicated as contributors to COVID-19 susceptibility and severity [114].

The association between EDCs and COVID-19 is likely mediated, in part, by the epigenetic regulation of key immune pathways involved in the host response to SARS-CoV-2.

## 12. Conclusions

Research on EDCs is very fertile but, nonetheless, little is yet known about this topic, despite 40 years having passed since the first study on the subject. It is a very important area of research, given its implications for the health of citizens, especially children since EDCs are influencing their development from an early age. From what is known, it seems that exposure to EDCs negatively impacts adults and children’s health through disruption of endocrine function; it seems that this effect is exerted by interfering at many levels with multiple endocrine and neurological patterns, so it is essential in this context to bear in mind what is already known about endocrine disruptors and to deepen our knowledge on this mechanism of interaction in order to adopt rules of conduct that allow limiting exposure to EDCs and their negative effects on human health [115]. Moreover, it is important to continue the research in order to recognize which, among the many products in this regard, are valid and not toxic; therefore, further studies are needed to obtain more robust data on the correlation between EDCs and the development of endocrine and neurological dysfunctions.

While research about EDCs is still working on assessing what is listed above, considering the complexity of the interferences and the not immediate effects, the task of pediatricians must be to give children and their families good cautious rules of behavior in order to minimize exposure to these substances. Considering the growing evidence that even low-dose human exposures can be harmful, clinicians should be aware and alert. The task of clinicians, therefore, is to work on both sides: to pursue a deepened knowledge of EDCs’ effects and mechanisms of action and to provide useful information; as an example, we cite the Italian Ministry of Environment, which in cooperation with the National Institute of Health, published a decalogue for the citizen entitled “Know, Reduce, Prevent Endocrine Disruptors” freely available on their website. The decalogue was born within the PREVIENI project [116] and the European guidelines on environmental risk [117] and aims to provide citizens and doctors in the territory with a comprehensive handful of information on the risks of daily environmental exposure to certain chemicals, in order to adopt conscious choices and behaviors that concern the general population and specifically the pediatric population.

Lastly, since the COVID-19 pandemic has been related to an increased spread of EDC exposure, it might be useful to monitor the consequences of this phenomenon during the next year; this might help us to achieve a deeper understanding of EDCs’ negative effects and the correlation between them and time/dose of exposure.

## Figures and Tables

**Figure 1 ijms-24-02671-f001:**
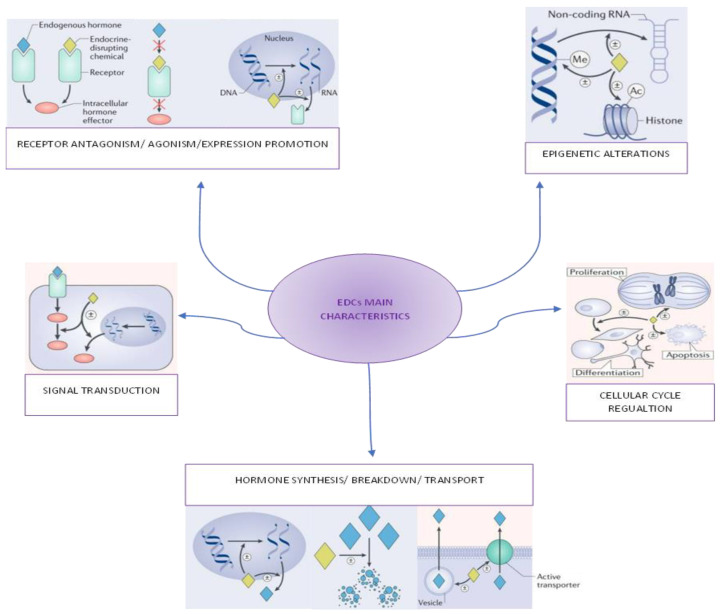
Main characteristics of EDCs. Adapted by Michele A La Merill et al., Nature Reviews 2020 [30]. The ± symbol indicates an EDC’s effect (increase/decrease processes).

## Data Availability

No new data were created or analyzed in this study. Data sharing is not applicable to this article.

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
