# Peer review of "Endocrine Disruptor Chemicals and Children’s Health"

_ijms, 2023, doi:10.3390/ijms24032671_

Round 1
Reviewer 1 Report
The review covers a very broad subject and it is unclear what is new in the text. Because the authors didn't focus on a specific endpoint, it says it all without details sometimes without direct references which would allow readers to go further in details . Important references are missing.
Table 1 and Figure 1 do not add anything.
Perhaps the authors should think more about the angle from which they want to approach the question of endocrine disruptors. What is the message to convey; and from there, propose a plan and roll it out, attaching the appropriate references.
Author Response
Point 1: The review covers a very broad subject and it is unclear what is new in the text. Because the authors didn't focus on a specific endpoint, it says it all without details sometimes without direct references which would allow readers to go further in details. Important references are missing.
Response 1: Thanks for your observation. Endocrine disruptors represent an extremely vast topic of which little is known, especially in the pediatric field, Our purpose is appropriate to cover the most recent information about the major EDCs (in terms of effects, routes of exposure and structure) in order to identify the most promising research fields. To achieve this goal, it was necessary to find a balance between going over the main aspects and not going into too much detail. We hope we have managed to clarify our goal. Please see the attachment.
Point 2: Table 1 and Figure 1 do not add anything.
Response 2: We agree that Figure 1 is too simple and unnecessary: we created it thinking we could provide an overview and an impact image on the organs concerned, but we agreed it should be removed. For what table 1 is concerned, instead, another reviewer found it appropriate, therefore, we would like to know if you have any suggestions on how to improve its function of summarize main EDCs routes of exposure and effects.
Point 3: Perhaps the authors should think more about the angle from which they want to approach the question of endocrine disruptors. What is the message to convey; and from there, propose a plan and roll it out, attaching the appropriate references.
Response 3: Thank you for your comment. We have tried to improve the paper calrifying our purpose in the introduction and conclusion. Please see the attachment.

Reviewer 2 Report
This is a review article dealing with endocrine-disrupting chemicals (EDCs) and their effect on children’s health.
After the introduction, the authors give us the general characteristics of endocrine disruptors, followed by a description of each one's unique properties, including how they enter the body, their mechanism of action, and possible adverse effects. The content is scientific and covers all areas of interest. The paper is written comprehensibly and has a sizable number of references. It contains one table and one figure that are well explained. Figure 1 is simple and not necessary because affected organs are mentioned in the text.
However, there are a lot of issues regarding citations and abbreviations. In parts 7, probably 8 (number 8 is missing), and 9 the references are given after the subheading and not in the text, which is inappropriate for a scientific article. Because the references cited in the text are not provided, it is hard to follow it properly. The cited reference must come after the paragraph that refers to it.
Many of the abbreviations that are used in the article are not defined. On the other hand, some abbreviations are explained in several places. Every one of the abbreviations used in the text, from beginning to conclusion, must be explained the first time it is used. This problem is mostly in parts 7-9.
It is clear that the article was written by at least two people because the content is not balanced and there is a clear difference in the writing of certain parts.
References are not listed in the manner required by the journal
In several references, complete data are not given.
It is not clear what reference 85 is, and why it is cited in the conclusion.
Author Response
Point 1: This is a review article dealing with endocrine-disrupting chemicals (EDCs) and their effect on children’s health.After the introduction, the authors give us the general characteristics of endocrine disruptors, followed by a description of each one's unique properties, including how they enter the body, their mechanism of action, and possible adverse effects. The content is scientific and covers all areas of interest. The paper is written comprehensibly and has a sizable number of references. It contains one table and one figure that are well explained.
Response 1: Thank you for your opinion and for your encouragement.
.
Point 2: Figure 1 is simple and not necessary because affected organs are mentioned in the text.
Response 2: Thank you for sharing your opinon. We agree with you and have decided to remove the figure, replacing it with something more useful. Please let us know if you have suggestions on a possible new figure or on how to improve that proposal.
Point 3: However, there are a lot of issues regarding citations and abbreviations. In parts 7, probably 8 (number 8 is missing), and 9 the references are given after the subheading and not in the text, which is inappropriate for a scientific article. Because the references cited in the text are not provided, it is hard to follow it properly. The cited reference must come after the paragraph that refers to it.
Response 3: We put your advice into practice. Please see the attachment.
Point 4: Many of the abbreviations that are used in the article are not defined. On the other hand, some abbreviations are explained in several places. Every one of the abbreviations used in the text, from beginning to conclusion, must be explained the first time it is used. This problem is mostly in parts 7-9.
Response 4: Thank you for your careful analysis. We corrected this specific aspect. Please see the attachment.
Point 5: It is clear that the article was written by at least two people because the content is not balanced and there is a clear difference in the writing of certain parts.
Response 5: Your observation is correct, we tried to balance our writing style (Please see the attachment), if you have any further suggestion on how to make the whole work more fluent it will be very handful, thank you in advance.
Point 6: References are not listed in the manner required by the journal. In several references, complete data are not given.
Response 6: We have revised the bibliography.
Point 7: It is not clear what reference 85 is, and why it is cited in the conclusion.
Response 7: We have corrected the citation in the bibliography to make it clearer. This is an article published in an Italian scientific journal where the impact of endocrine disruptors on children's health is recognized and some practical and useful advice is provided. In the conclusions, we reiterate the importance of accurate research on endocrine disruptors for its relevance in clinical practice. The possibility of transforming knowledge on endocrine disruptors into practical advice emerges from the cited article.

Reviewer 3 Report
The manuscript entitled "Endocrine Disruptors Chemicals and Children’s Health", is really very interesting, the topic is actual and with high relevance. In my opinion this type of revision is really important and stablish the point that must be studied in future research.
I think that this manuscript could be accepted after major revision.
Line 80-106. Please, add references for example, Bujalance et al., 2022 https://doi.org/10.3390/ani12030300 (these authors exposed mice to BPA for successive generations, and observed variations in glucose, lipids, albumin, and other biochemical parameters), there are several manuscripts related to this parameters evaluation.
Line 118-120- The authors must refer to that regulation in which the use was banished in Europe in 2011 in food packaging. As far as I know, the use of BPA is restricted in Europe but not prohibited (European Reglament 2018/213)
Lines 126-131. Please, add references
Lines 145-156. Please, add references
Table 1. I can´t understand it. What does the possible actions column refer to? Because in such a case there are no possible effects? What does the column indicating absorption mechanisms refer to? Absorption pathways? The term mechanism here may not be correct. The absorption pathways are mixed with concepts such as bioaccumulation or crossing the placental barrier. You should clarify these concepts, and add another column if it proceed. Does BPA not cross the placental barrier? Review well the content of this table and indicate with bibliographic references.
Figure 1. I can´t understand this figure. The main affected organs are different in males than females? Obviously the reproductive systems is different, but the rest of the organs? Why do the authors mark part in the man and part of them in the woman figure?
Line 440. Please check if citation 58 is related to the BPA ban, as indicated in that line.
Author Response
Point 1: The manuscript entitled "Endocrine Disruptors Chemicals and Children’s Health", is really very interesting, the topic is actual and with high relevance. In my opinion this type of revision is really important and stablish the point that must be studied in future research.
I think that this manuscript could be accepted after major revision.
Response 1: thank you for your opinion and for your encouragement.
Point 2: Line 80-106. Please, add references for example, Bujalance et al., 2022 https://doi.org/10.3390/ani12030300 (these authors exposed mice to BPA for successive generations, and observed variations in glucose, lipids, albumin, and other biochemical parameters), there are several manuscripts related to this parameters evaluation.
Response 2: We addedd more references, we avoided on purpose to cite articles about mice because we wanted to limit our observations to the human population, if you still think this citations are necessary we will proceed in adding them.
Point 3: Line 118-120- The authors must refer to that regulation in which the use was banished in Europe in 2011 in food packaging. As far as I know, the use of BPA is restricted in Europe but not prohibited (European Reglament 2018/213)
Response 3: Thank you for your clarification, we managed to modify this point.
Point 4: Please, add references. (Lines 126-131 and 145-156).
Response 4: We have added references. Thank you for your suggestion.
Point 5: Table 1. I can´t understand it. What does the possible actions column refer to? Because in such a case there are no possible effects? What does the column indicating absorption mechanisms refer to? Absorption pathways? The term mechanism here may not be correct. The absorption pathways are mixed with concepts such as bioaccumulation or crossing the placental barrier. You should clarify these concepts, and add another column if it proceed. Does BPA not cross the placental barrier? Review well the content of this table and indicate with bibliographic references.
Point 5: Thanks for your comment. We agree on the need to modify the table, we hope that with these new revisions the issue is fixed.
Point 6: Figure 1. I can´t understand this figure. The main affected organs are different in males than females? Obviously the reproductive systems is different, but the rest of the organs? Why do the authors mark part in the man and part of them in the woman figure?
Response 6: A clear gender difference in the action of endocrine disruptors has not been demonstrated. We have created the image in order to provide a simple and immediate visual exemplification of the organs involved. Since the figure adds no information to the text and may even lead to doubts about the matter, we have decided to remove it and replace it with a different image which we hope will result more handful. Thank you for your comment.
Point 7: Line 440. Please check if citation 58 is related to the BPA ban, as indicated in that line.
Response 7: Pleases see the attachment.

Round 2
Reviewer 1 Report
Sorry but I cannot see modifications in the text. They should be underlined or highlight in red or any color
I will not compare the text with the previous one line by line
Author Response
Dear rewier, in the attachments we highlighted in red the section in the introduction and cocnlusion that we think will be more useful in understanding the purpose of the article as required in the first report.
As said in the 1st report, we also modified the image and the table as well as their descriptions.

Reviewer 2 Report
In the new version of the article the abbreviations are explained, the picture is replaced and chapter 9 is corrected. But there is still a serious deficiency reflected in the absence of the cited references in chapters 7 and 8 making them not sound like a part of a scientific article.
For example:
line 268-270 – please provide a reference
line 272-274- for the statement made, please provide a reference.
Line 274- please, provide a reference for mentioned meta-analysis
Line 278 – please, provide a reference
Line 283-287 please, provide a reference
etc
Only references that are cited in the article's text may be included in the list of references. In the updated version of the article twenty additional references that are not mentioned in the text have been added to the list of references.
References are still not listed as required in the journal (Instructions for authors)
Author Response
Dear rewier, thanks for your observations, we tried to adjust the manuscript following your suggestions
1st issue: there is still a serious deficiency reflected in the absence of the cited references in chapters 7 and 8
Answer 1 : we addedd the required references.
2nd issue: Only references that are cited in the article's text may be included in the list of references. In the updated version of the article twenty additional references that are not mentioned in the text have been added to the list of references.
References are still not listed as required in the journal
Answer 2: the references that have been added are meant to clarify the sources of our statement as required in the 1st report and in this second report.
We modified the references list as required.

Reviewer 3 Report
The manuscript has been modified following my suggestions, the figure and table have been modified too. Once I have reviewed the manuscript I think that is more clear and in my opinion can be accepted for publication.
Congratulations
Author Response
Dear rewier thank you for your support and your appreciation
Round 3
Reviewer 1 Report
The ms has improved. I have comments regarding references.
line 142: reference 6 must be an error. Please include references with PMID 30333793 and PMID 18174957
line 162: include reference with PMID 35714525 and discuss dosage of pollutants in hair versus urine
from line 177: include reference wih PMID 29426018
line 278: present data of reference with PMID 35461094
from line290: on reading the text, I understand that all endocrine disruptors act by all the different modes of action presented, which is not the case. Please specify which EDCs or groups of EDCs were found to be deregulate peripheral IR or others
line 318: chemical molecules are not capitalized
Author Response
POINT 1
line 142: reference 6 must be an error. Please include references with PMID 30333793 and PMID 18174957
line 162: include reference with PMID 35714525 and discuss dosage of pollutants in hair versus urine
from line 177: include reference wih PMID 29426018
line 278: present data of reference with PMID 35461094
RESPONSE 1: We included the references and we corrected as suggested.
POINT 2: from line290 - on reading the text, I understand that all endocrine disruptors act by all the different modes of action presented, which is not the case. Please specify which EDCs or groups of EDCs were found to be deregulate peripheral IR or others.
RESPONSE 2: For what this point is concerned we referred generically to the main EDC listed above collecting all their main effects because we wanted to give a general insight on this aspect. Since it is not the only issue analyzed in the paper we thought that specifying the multiple and complex effects for every single EDC would have meant writing a paragraph that would have been too long and specific; nevertheless we added some clarifications and quoted the article (30) in which every detail is better explained.
POINT 3: line 318 - chemical molecules are not capitalized
RESPONSE 3: thanks for your suggestion. Chemical molecules are now capitalized.

Reviewer 2 Report
Unfortunately, I think that the given corrections are not sufficient for the manuscript to be published in this journal. The manuscript is well written, but the same issue persists in chapters 7-9 with references not provided in the appropriate places or listed in the order they appear in the text. Following reference 31, there are references 108, 110, 32-33,...
Author Response
Point 1: references are not provided in the appropriate places or listed in the order they appear in the text.
Answer 1: Thanks for your comments. We have made the required corrections. We hope that the manuscript is now properly edited.
